

# Residual dynamics and dietary exposure risk of dimethoate and its metabolite in greenhouse celery

Chunjing Guo[1,2], Guang Li[1,2], Qiujun Lin[1,2], Xianxin Wu[1,2] and Jianzhong Wang[1,2]

[1] Institute of Agricultural Quality Standards and Testing Technology, Liaoning Academy of Agricultural Sciences, Shenyang, China
[2] Ministry of Agriculture and Rural Affairs Lab of Agricultural Product Quality Safety Risk Assessment (Shenyang), Shenyang, China

Corresponding author
Jianzhong Wang,
WJZ721125@sina.com

## ABSTRACT

This study aimed to explore the residual dynamics and dietary risk of dimethoate and its metabolite omethoate in celery. Celery was sprayed with 40% dimethoate emulsifiable concentrate (EC) at either a low concentration of 600 g a.i./ha or a high concentration of 900 g a.i./ha. Plants in the seedling, transplanting, or middle growth stages were sprayed once, and the samples were collected 90 days after transplantation. Plants in the harvesting stage were sprayed two or three times. The samples were collected on days 3, 5, 7, 10, 14 and 21 after the last pesticide application. The dimethoate and omethoate compounds were extracted from the celery samples using acetonitrile, and their concentrations were detected using ultra-performance liquid chromatography-tandem mass spectrometry. Also, the dietary risk assessments of dimethoate and omethoate were conducted in various populations and on different foods in China. The metabolism led to the formation of omethoate from dimethoate in the celery. The degradation dynamics of dimethoate and total residues in greenhouse celery followed the first-order kinetic equation. The half-lives of the compounds were 2.42 days and 2.92 days, respectively. The celery which received one application during the harvesting stage had a final residue of dimethoate after 14 days, which was lower than the maximum residue limit (MRL) 0.5 mg kg$^{-1}$ for Chinese celery. The final deposition of the metabolite omethoate after 28 days was less than the maximum residue limit of 0.02 mg kg$^{-1}$ for Chinese celery. Furthermore, the risk quotients of dimethoate in celery were less than 1; therefore, the level of chronic risk was acceptable after day 21. Only children aged 2–7 years had an HQ of dimethoate more than 1 (an unacceptable level of acute risk), while the acute dietary risks to other populations were within acceptable levels. It was recommended that any dimethoate applications to celery in greenhouses should happen before the celery reached the harvesting stage, with a safety interval of 28 days.

## INTRODUCTION

Celery is a good source of vitamin C, folic acid, carotene, phenols, and flavonoids (*Liang et al., 2018*), which are known to lower blood pressure (*Madhavi et al., 2013*) and have

**Figure 1 Chemical structure of dimethoate (A) and omethoate (B).**

anti-inflammatory and antioxidant effects in humans and other mammals (*Kooti & Daraei, 2017*; *Powanda & Rainsford, 2011*). China ranks first in the world in terms of celery production with a planting area of around 550,000 ha (*Gao et al., 2014*; *Madhavi et al., 2013*). Pesticides are commonly used in celery production to increase crop yield and quality by preventing and reducing the damage caused by diseases and insect pests. It is safe to use acetamiprid, thiamethoxam, imidacloprid, and pymetrozine in celery production according to the Pesticide Fact Sheet. However, the application of dimethoate has potential risks to the environment and human health; therefore, the risk assessments of pesticide residues have received increasing attention (*Dai et al., 2019*; *Dominiak, 2019*; *Kranawetvogl et al., 2018*; *Rezaei & Mahdi, 2018*).

Dimethoate (Fig. 1A), a broad-spectrum systemic insecticide is widely used to control insect pests in vegetables, fruits, tea trees, wheat and rice (*Zheng & Sun, 2014*). It can be degraded into omethoate (Fig. 1B) in plants. Omethoate is a highly toxic pesticide with strong contact and penetration effects (*Eddleston et al., 2006*; *Zhang et al., 2017*), and hence it has been banned from use on vegetables in China. However, recent investigations found that the detection rate and over standard rate of this pesticide residues are relatively high in celery (*Liu, 2017*; *Sun et al., 2014*; *Yaojun, 2016*). It was believed that the primary cause of this problem might be the production of omethoate as a metabolite when celery was sprayed with dimethoate. Its toxicity is much higher.

In China, the registration pre-harvest interval (PHI) of dimethoate in wheat is 14 days, but it is not registered in celery. Since omethoate derives from dimethoate, even if the safety interval of 14 days is safe for dimethoate, it does not mean that the metabolite omethoate is safe. Whether the current safety interval for dimethoate application ensures the residue of its metabolite omethoate to be below the maximum residue limit (MRL) is not clear. A previous study showed that the safety interval for celery sprayed with dimethoate should be 21 days for the residue of omethoate to be less than the MRL (*Guo et al., 2017*). Nevertheless, the demand for fresh vegetables in winter increases with the improvement in living standards, resulting in the expansion of the celery plantation area in northern greenhouses and greater use of dimethoate. Therefore, it is vital to monitor dimethoate and omethoate residues in greenhouses to assess human health risks (*European Food Safety Authority, 2016*; *Van, Pennell & Zhang, 2016*; *Zhu et al., 2015*).

Although efforts have been put into studying the dynamics of dimethoate in celery (*Chen et al., 2018*; *Lu et al., 2017*; *Yuan et al., 2014*), few reports are available on the degradation dynamics of omethoate residues. This study investigated the dissipation dynamics and residues of dimethoate and omethoate in greenhouse celery. Based on the experimental data, dietary risk assessments were conducted for different populations in China, and the safe application of dimethoate in celery was explored.

## MATERIALS AND METHODS

### Test materials

Celery was used as the test crop. The field test was carried out in the vegetable production base in the Liaozhong district of Shenyang City. No extreme weather events, such as heavy rain and hail, were observed during pesticide applications, and the climatic conditions were normal. The test pesticide was 40% dimethoate EC, Hebei Zhongtian Bangzheng Biologic Science Co., Ltd.. The maximum recommended dose is 600 g a.i./ha in China. The formulation of dimethoate was analyzed before application in this study. The content of dimethoate met the requirements, and no omethoate was detected in it.

### Instruments

Waters UPLC TQ Ultra Performance Liquid Chromatography-Tandem Mass Spectrometer, Waters, USA; Zhongjia HC 3514 High-Speed Centrifuge, Anhui USTC Zonkia Scientific Instrument Co., Ltd.; Ding Haoyuan RS-1 Vortex Mixer, Beijing Ding-Hao Yuan Technology Co., Ltd.; Pine-tree ultra-pure water machine, Beijing Xiangshunyuan Technology Co., Ltd.; JACTO HD400 Backpack Sprayer, JACTO Agricultural Machinery Co., Ltd.; 0.22 μm needle filter, 50 mL polypropylene plastic centrifuge tube, Xinkang Medical Equipment Co., Ltd.

### Reagents

Methanol, acetonitrile (chromatographically pure), Merck. Wondapak QuEchERS extraction and separation kit, Shimadzu Kojima (Shanghai) Trading Co., Ltd. Dimethoate and omethoate were purchased from the Environmental Quality Supervision and Testing Center of the Ministry of Agriculture (Tianjin).

### Standard solutions

Standard stock solutions (100 mg $L^{-1}$) of dimethoate and omethoate were diluted with acetonitrile to make the working standard solution of different concentrations (0.005, 0.01, 0.02, 0.05, 0.1, 0.2 and 0.5 mg $L^{-1}$). Additionally, celery samples cultivated in control plots were used as blanks. The stock solution was diluted with the clean control extract to generate the matrix standard solution (0.005, 0.01, 0.02, 0.05, 0.1, 0.2 and 0.5 mg $L^{-1}$). Standard solutions were stored in the dark at −20 °C. Blanks with a dimethoate and omethoate solution at three concentration levels (0.01, 0.1 and 1 mg $kg^{-1}$) were employed for the recovery assay. The analytical method's performance parameters, such as linear ranges, limit of quantification (LOQ), and limit of detection (LOD), were determined in addition to recovery rates.

## Field test design

According to Guideline's requirements on pesticide residues trials (*NY/T 788-2004, 2004*), the test plot was designed with a plot area of 30 m$^2$, a buffer zone of 2 m, and three repeat plots, which were randomly arranged. A control area of 30 m$^2$ without pesticide application was also set up to collect control samples.

### Dissipation dynamics

Dimethoate was sprayed at 900 g a.i./ha (1.5 times the maximum recommended dose) using a knapsack sprayer on the surface of celery in the middle growth stage, and the experiment was repeated on three plots. The samples were collected two hours, 1, 3, 5, 7, 10, 14, 21, 28 and 42 days after pesticide application.

Final residual dynamics: The pesticide application dose was 600 g a.i./ha (the maximum recommended dose) and 900 g a.i./ha (1.5 times the maximum recommended dose), respectively. Dimethoate was sprayed once using a knapsack sprayer on the soil in the seedling stage, and ripe celery samples were collected 145 days after the pesticide application. Dimethoate was sprayed once using a knapsack sprayer on the surface of celery in the transplanting stage, and ripe celery samples were collected at 90 days after the pesticide application. Dimethoate was sprayed once using a knapsack sprayer on the celery surface in the middle growth stage, and samples of ripe celery were collected at 45 days after the pesticide application. Besides, dimethoate was sprayed using a knapsack sprayer on the surface of celery two and three times in the harvesting stage with intervals of 7 days between applications, and the experiment was repeated on three plots. The samples were collected at 3, 5, 7, 10, 14 and 21 days after the last pesticide application.

The seedling stage was the day of sowing, the transplanting stage was 55 days after sowing, while the middle growth stage was 45 days after transplantation. Finally, ripe celery was collected 90 days after transplantation. The harvesting stage was 62–97 days after transplantation. The samples were collected 3, 5, 7, 10, 14 and 21 days after the last pesticide application. The growing stage, days of the pesticide application and sampling are shown in Fig. 2.

### Sampling

For this, 2 kg of standard, damage-free celery samples 2 cm above the ground were randomly collected from 5 to 12 points in each plot each time. No samples were collected within 0.5 m of the edge of the field. The samples were placed in polyethylene bags and transported to the laboratory for the next study stage. The samples were homogenized using a blender (Foer Group, Hong Kong Special Administrative Region, China) and stored in a refrigerator at −18 °C until use.

## Sample analysis

### Extraction

First, 10.0 g of the sample to be tested was weighed and placed in a 50 mL centrifuge tube. Second, 20.0 mL of acetonitrile was added to the centrifuge tube and homogenized for 2 min. The QuEChERS extraction separation bag was added with vigorously shaking for
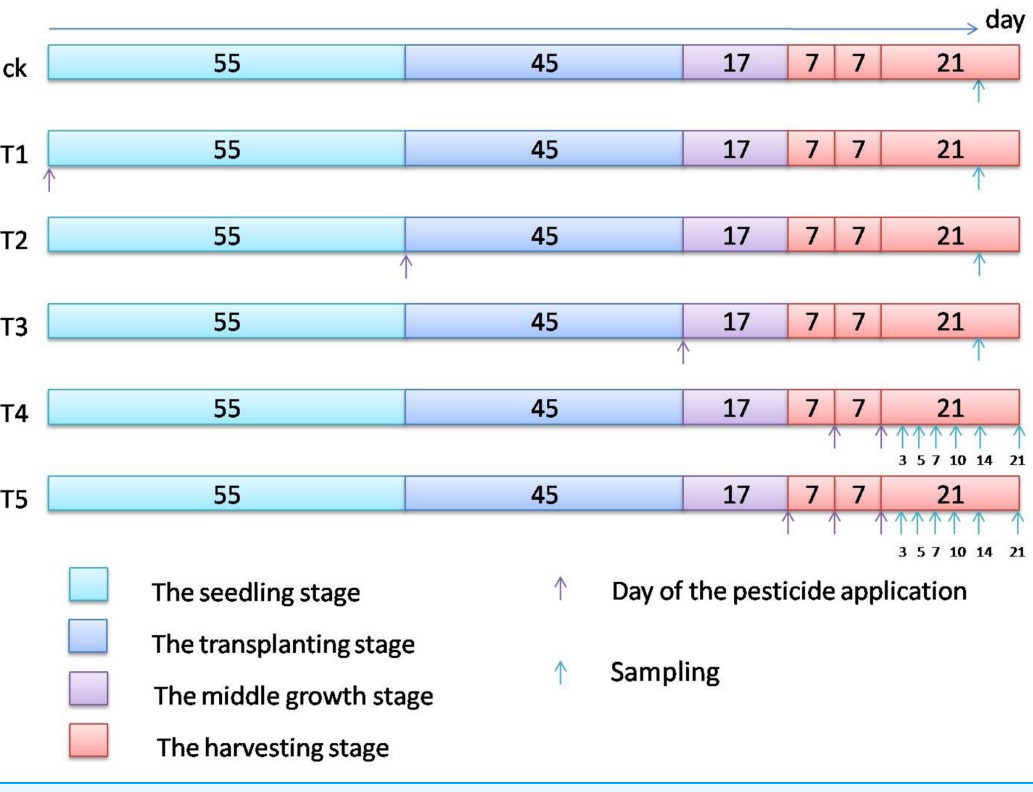

Figure 2 **Growing stage, day of the pesticide application and sampling.**

2 min and centrifuged at 10,000 rpm for 5 min. Finally, the solution supernatant was filtered with a 0.22-μm filter membrane, and the filtrate was ready to be tested.

## Detection

Chromatographic conditions were as follows: Acquity UPLC HSS T3 column (100 mm × 2.1 mm, 1.8 μm); column temperature, 25 °C; injection volume, 5 μL; flow rate, 0.38 mL min$^{-1}$; mobile phase A, water, and mobile phase B, methanol. Gradient elution conditions were as follows: 0–0.25 min, 90–5% A; 0.25–5.00 min, 5–90% A. Mass spectrometry conditions were as follows: electrospray ion source positive ion mode (ESI +); ion source temperature, 500 °C; capillary voltage, 1.0 kV; nebulizing gas flow rate, 900 L h$^{-1}$; and taper hole anti blow air flow rate, 50 L h$^{-1}$. The scanning method was the multiple reaction monitoring (MRM) mode. The other MS/MS parameters were separately optimized for each target compound and are listed in Table 1.

## Dissipation dynamics

The first-order kinetic equation was used to express the dissolution dynamics of dimethoate and omethoate in celery over time.

$$c_t = c_0 e^{-kt} \tag{1}$$

**Table 1 Details of tandem mass spectrometry parameters of dimethoate and omethoate.**

| Pesticide | Precursor ion, m/z | Product ion, m/z | Collision energy, eV | Declustering potential, V |
|---|---|---|---|---|
| Dimethoate | 230.0 | 125.2 | 12 | 20 |
| | 230.0 | 199.3 | 12 | 10 |
| Omethoate | 214.4 | 125.1 | 15 | 20 |
| | 214.4 | 183.2 | 15 | 11 |

$$t_{1/2} = \frac{Ln2}{k} \tag{2}$$

where $t$ is time (day), $c_t$ is the concentration (mg kg$^{-1}$) at time t (days), $c_0$ is the initial concentration (mg kg$^{-1}$), $k$ is the degradation rate constant (day$^{-1}$) and $t_{1/2}$ is the half-life (d).

## Final residue

The toxicological endpoints of dimethoate and its metabolite are the same; therefore, the sum of residues of dimethoate and omethoate should be considered together for both acute and chronic dietary intake. Omethoate is more toxic than dimethoate. The relative toxicity of omethoate compared to dimethoate following chronic and acute were found to be about ~3:1 and ~6:1, respectively (*None, 2009*).

Sum of dimethoate and 6×omethoate, expressed as dimethoate (for acute risk assessment);

Sum of dimethoate and 3×omethoate, expressed as dimethoate (for chronic risk assessment).

A World Health Organization (WHO) template for evaluating acute exposure (IESTI) is used to evaluate the risk of acute dietary exposure uses. In contrast, a WHO template to evaluate chronic exposure (IEDI) is used to evaluate the risk of chronic dietary exposure. (http://www.who.int/foodsafety/areas_work/chemical-risks/gems-food/en/).

The following formula (*Geng et al., 2018*) was used to calculate the risk of chronic dietary exposure of dimethoate and omethoate.

$$NEDI = F \times STMR/bw \tag{3}$$

$$RQ = NEDI/ADI \tag{4}$$

where *NEDI* is the country's estimated daily intake (mg kg$^{-1}$ bw day$^{-1}$), *STMR* is the median residue of the standard test (mg kg$^{-1}$), *F* is the average food consumption (kg d$^{-1}$), *bw* is the body weight (kg) and *ADI* is the acceptable daily intake (mg kg$^{-1}$ bw day$^{-1}$). *RQ* is chronic risk assessment. *RQ* > 1 indicates that the chronic dietary intake risk is unacceptable. *RQ* < 1 suggests that the chronic nutritional intake risk is acceptable; the smaller the risk quotient (RQ), the lower the risk.

The following formula was used to calculate the risk of acute dietary exposure of dimethoate (the single weight of unprocessed food was more than 25 gs and the edible

**Table 2 Linear regression parameters of the calibration curve for dimethoate and omethoate in pure solvent and matrices.**

| Compounds | Matrix | Range (mg L$^{-1}$) | Regression equation | R | Slope ratio | ME (%) |
|---|---|---|---|---|---|---|
| Dimethoate | acetonitrile | 0.005–500 | $y = 146.345x+41.4$ | 0.9992 | – | – |
| | Celery* | 0.005–500 | $y = 141.177x+226$ | 0.9993 | 0.96 | −4 |
| Omethoate | acetonitrile | 0.005–500 | $y = 35.019x+64.79$ | 0.9986 | – | – |
| | Celery* | 0.005–500 | $y = 33.5158x+172$ | 0.9960 | 0.96 | −4 |

Note:
* Celery samples cultivated in control plots were used as blanks. The stock solution was diluted with the clean control extract to generate the matrix standard solution.

portion's available weight was more than or equal to the consumption by most individuals) (*Geng et al., 2018*).

$$IESTI = LP \times HR \times v/bw \quad (5)$$

$$HQ = IESTI/ARfD \quad (6)$$

where *IESTI* is the estimated short-term intake (mg kg$^{-1}$ bw day$^{-1}$), *LP* is the average food consumption (kg d$^{-1}$), *HR* is the highest residue obtained in the test (mg kg$^{-1}$), *v* is the variability factor assigned a value of 3 according to JMPR (*Gao, Chen & Zhang, 2007*), *bw* is the body weight (kg), and *ARfD* is the acute reference dose (mg kg$^{-1}$ bw day$^{-1}$). *HQ* is an acute risk assessment. When *HQ* < 1, which means that the risk of acute dietary intake is acceptable. When *HQ* > 1, it indicates an unacceptable acute risk.

## RESULTS

### Method validation

The limits of detection (LODs) and the limits of quantification (LOQs) for dimethoate and omethoate were considered to be the concentrations produced at a signal-to-noise (S/N) ratio of 3 and 10, respectively. The LODs for the two target chemicals were 0.003 mg kg$^{-1}$, and the LOQs were 0.01 mg kg$^{-1}$. Good linear calibration curves were obtained over the concentration range of 0.005–0.5 mg L$^{-1}$ for dimethoate and omethoate and the correlation coefficient r was higher than 0.99 (Table 2). The sample concentrations outside the linear range were diluted to the appropriate analytical concentration. The matrix effect (ME) was calculated:

$$ME(\%) = (slope_{ratio}-1) \times 100\% \quad (7)$$

$$slope_{ratio} = slope_{matrix}/slope_{solvent} \quad (8)$$

where slope matrix and slope solvent are the calibration curve slopes of the celery and acetonitrile standard, respectively. The matrix effects (MEs) were −4% (Table 2), which caused the signal's suppression. Thus, matrix-matched calibration solutions were used to compensate for errors associated with matrix-induced calibration.

The accuracy was evaluated by determining the recovery assay at three levels in celery. No dimethoate and omethoate were detected in the blanks. The mean recoveries were

**Table 3 Recoveries and relative standard deviations (RSDs) of dimethoate, and omethoate in celery at different fortification levels ($n$ = 6).**

| Pesticide | Fortification (mg kg$^{-1}$) | Celery | |
|---|---|---|---|
| | | Mean recovery (%) | RSD (%) |
| Dimethoate | 0.01 | 83.4 | 3.7 |
| | 0.1 | 86.6 | 4.5 |
| | 1 | 92.9 | 4.0 |
| Omethoate | 0.01 | 80.4 | 4.0 |
| | 0.1 | 88.8 | 7.2 |
| | 1 | 94.6 | 7.3 |

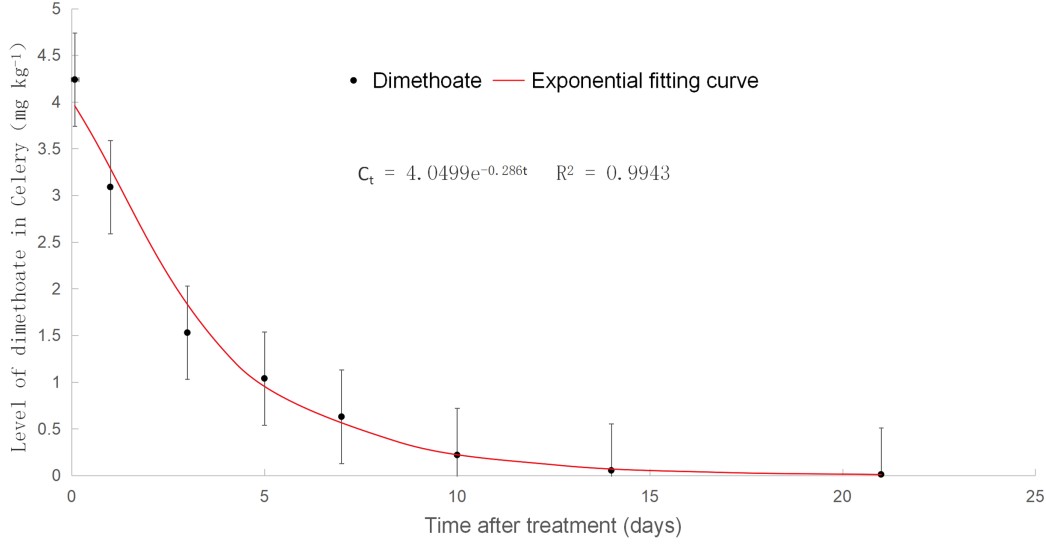

**Figure 3 Level of dimethoate in celery.**           

83.4–92.9% and 80.4–94.6% for dimethoate and omethoate, with RSD in the range of 3.7–4.5% and 4.0–7.3%, respectively (Table 3). This finding indicated that the method of analysis was accurate and precise.

## Dimethoate dissipation dynamics in celery

The results of dimethoate detection were expressed as average values of three repeat plots. As shown in Fig. 3 (when the safety interval was more than 28 days, and the concentration of dimethoate was lower than the LOQ), the degradation of dimethoate met the first-order kinetic equation, $C_t = 4.0499e^{-0.286t}$, and the correlation coefficient $r^2$ was 0.9943. The half-life was 2.42 days, indicating that dimethoate was an easily degradable pesticide. Ten days later, the dissipation rate reached 94.6%, and the residual concentration of dimethoate decreased to less than 0.5 mg kg$^{-1}$ (the MRL of dimethoate on celery is 0.5 mg kg$^{-1}$), which was lower than the MRL. Furthermore, the dissipation rate reached 99% after 16.1 days.

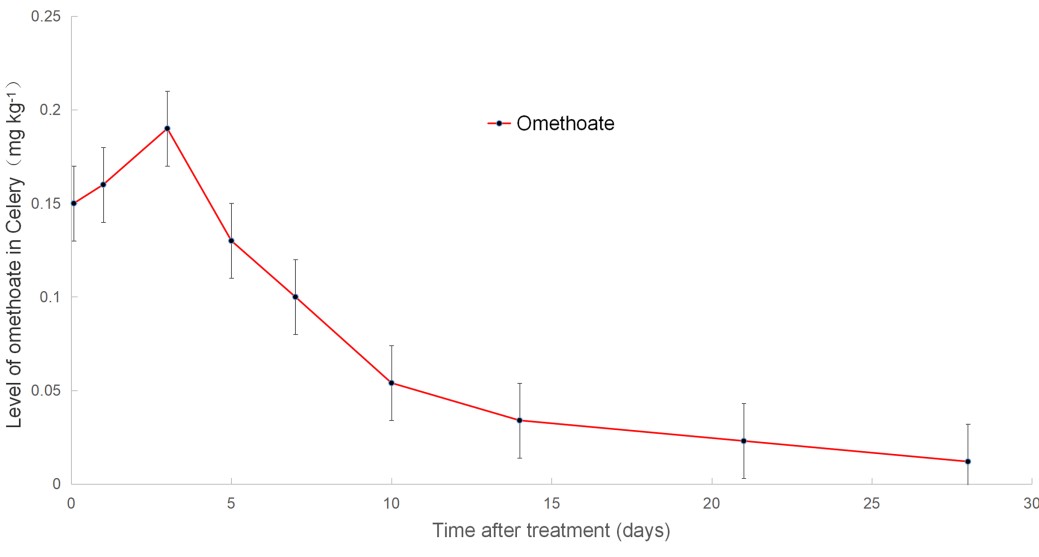

**Figure 4  Level of omethoate in celery.**               

## Omethoate dissipation dynamics in celery

The results of omethoate detection were expressed as average values of three repeat plots. The dissipation data fitting is shown in Fig. 4 (when the safety interval was more than 42 days, the concentration of omethoate was lower than the LOQ). Before application, the formulation of dimethoate was analyzed, and no omethoate was detected in it. However, omethoate was detected in the celery. After the application of dimethoate, the concentration of omethoate increased to 0.19 mg kg$^{-1}$ on day 3 and gradually decreased after 3 days. This finding indicated that the levels of omethoate present in the celery were high due to dimethoate metabolism. After 28 days, the omethoate concentration was less than 0.02 mg kg$^{-1}$, which was lower than the allowable MRL of omethoate in celery. Hence, a 10 day safety interval was sufficient to ensure that the dimethoate concentration in celery declined to safe levels but was not enough for the omethoate concentration to reach safe levels. A safety interval of 28 days after dimethoate application is recommended based on the MRL (0.02 mg kg$^{-1}$) of omethoate in Chinese celery.

As shown in Fig. 5, the dissipation behavior of total residues of dimethoate and its metabolite omethoate conformed to the first-order kinetic equation, $C_t = 3.7599e^{-0.237t}$, the correlation coefficient $r^2$ was 0.9814. The half-life was 2.92 days, which was 20.1% longer than that of parent compound dimethoate. This finding indicated that omethoate should be taken into account in risk assessment as a metabolite of dimethoate.

## Final residues following pesticide treatments in seedling, transplanting, and middle growth stages

The final residues of dimethoate and its metabolite omethoate after application in the seedling, transplanting, and middle growth stages of celery are shown in Table 4. The data showed that both residues of dimethoate and omethoate in celery were lower than the LOQ

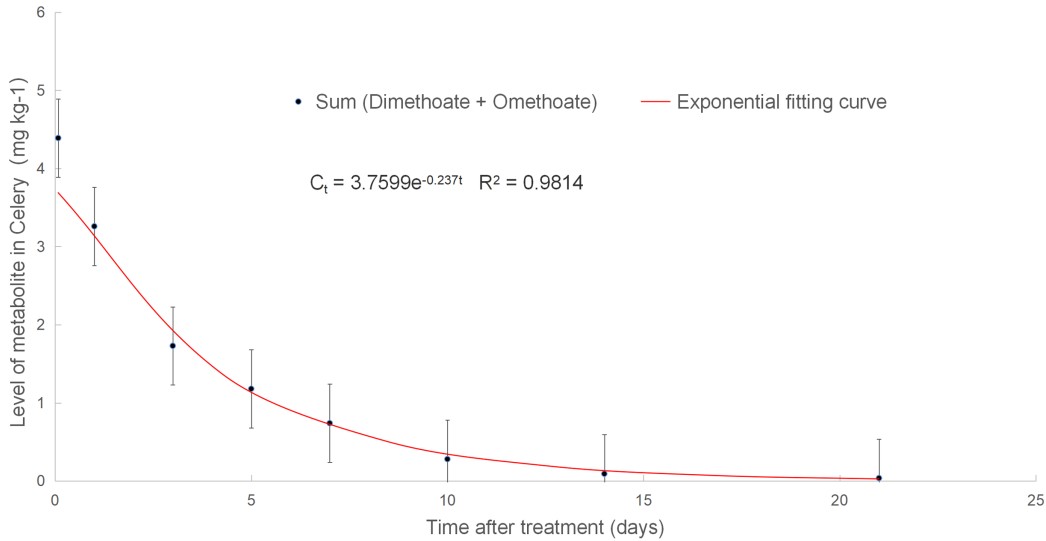

**Figure 5 Level of dimethoate and its metabolite in celery.**

**Table 4 Residues of dimethoate and its metabolite omethoate after application during seedling stage, transplanting stage and middle stage of growth the celery[*].**

| Pesticide | Dosage (g a.i./ha) | Final residue during seedling stage (mg kg$^{-1}$) | Final residue during transplanting stage (mg kg$^{-1}$) | Final residue during middle stage of growth (mg kg$^{-1}$) |
|---|---|---|---|---|
| Dimethoate | 600 | <0.01 | <0.01 | <0.01 |
| | 900 | <0.01 | <0.01 | <0.01 |
| Omethoate | 600 | <0.01 | <0.01 | <0.01 |
| | 900 | <0.01 | <0.01 | <0.01 |

**Note:**
[*] Seedling stage (145 days), transplanting stage (90 days) and middle stage of growth the celery (45 days).

and the MRLs (the MRL of dimethoate is 0.5 mg kg$^{-1}$, and the MRL of omethoate is 0.02 mg kg$^{-1}$).

## Final residues following pesticide treatments in the harvesting stage

The samples were collected at 3, 5, 7, 10, 14 and 21 days after the last pesticide application. The final residues of dimethoate and omethoate are shown in Tables 5 and 6. The data showed that, the residue of dimethoate in celery was lower than the allowable MRL of 0.5 mg kg$^{-1}$ after 10 days when two different dosages of dimethoate were sprayed two or three times. However, omethoate concentration was still higher than the allowable MRL of 0.02 mg kg$^{-1}$ after 21 days. Additionally, celery sprayed three times with the same dimethoate concentration had higher residues of dimethoate and omethoate compared with celery sprayed only twice, showing a cumulative effect of repeated pesticide application. However, whether the concentration of omethoate had declined to a level below the MRL by this stage could not be determined because a sample of celery 28 days after the final pesticide application was not collected.

**Table 5 Residues of dimethoate and its metabolite omethoate after the application of dimethoate two times during harvesting stage of growth the celery.**

| Pesticide | Dosage (g a.i./ha) | Final residue (mg kg$^{-1}$) (Days after the last application) | | | | | |
|---|---|---|---|---|---|---|---|
| | | 3 | 5 | 7 | 10 | 14 | 21 |
| Dimethoate | 600 | 1.60 | 1.15 | 0.78 | 0.35 | 0.064 | 0.014 |
| | | 1.61 | 1.16 | 0.74 | 0.33 | 0.058 | 0.015 |
| | | 1.58 | 1.13 | 0.75 | 0.35 | 0.062 | 0.018 |
| | RSD (%) | 1.5 | 1.5 | 2.1 | 1.2 | 0.31 | 0.21 |
| | 900 | 1.87 | 1.26 | 0.87 | 0.35 | 0.081 | 0.023 |
| | | 1.86 | 1.23 | 0.81 | 0.35 | 0.082 | 0.022 |
| | | 1.89 | 1.31 | 0.92 | 0.37 | 0.091 | 0.024 |
| | RSD (%) | 1.5 | 4.0 | 5.5 | 1.2 | 0.55 | 0.1 |
| Omethoate | 600 | 0.19 | 0.15 | 0.13 | 0.11 | 0.041 | 0.031 |
| | | 0.18 | 0.15 | 0.12 | 0.11 | 0.043 | 0.032 |
| | | 0.20 | 0.14 | 0.13 | 0.12 | 0.039 | 0.028 |
| | RSD (%) | 1.0 | 0.58 | 0.58 | 0.58 | 0.20 | 0.21 |
| | 900 | 0.21 | 0.16 | 0.14 | 0.10 | 0.058 | 0.041 |
| | | 0.20 | 0.16 | 0.14 | 0.11 | 0.054 | 0.041 |
| | | 0.22 | 0.17 | 0.13 | 0.15 | 0.059 | 0.046 |
| | RSD (%) | 1.0 | 0.58 | 0.58 | 2.6 | 0.26 | 0.29 |

**Table 6 Residues of dimethoate and its metabolite omethoate after the application of dimethoate three times during harvesting stage of growth the celery.**

| Pesticide | Dosage (g a.i./ha) | Final residue (mg kg$^{-1}$) (Days after the last application) | | | | | |
|---|---|---|---|---|---|---|---|
| | | 3 | 5 | 7 | 10 | 14 | 21 |
| Dimethoate | 600 | 1.73 | 1.21 | 0.82 | 0.41 | 0.074 | 0.035 |
| | | 1.73 | 1.22 | 0.80 | 0.43 | 0.075 | 0.031 |
| | | 1.72 | 1.21 | 0.83 | 0.40 | 0.078 | 0.036 |
| | RSD (%) | 0.58 | 0.58 | 1.5 | 1.5 | 0.21 | 0.26 |
| | 900 | 2.01 | 1.39 | 0.89 | 0.46 | 0.11 | 0.062 |
| | | 1.98 | 1.39 | 0.91 | 0.47 | 0.12 | 0.063 |
| | | 2.05 | 1.38 | 0.87 | 0.44 | 0.15 | 0.058 |
| | RSD (%) | 3.5 | 0.58 | 2.0 | 1.5 | 2.1 | 0.26 |
| Omethoate | 600 | 0.21 | 0.19 | 0.14 | 0.11 | 0.062 | 0.033 |
| | | 0.21 | 0.19 | 0.15 | 0.10 | 0.056 | 0.042 |
| | | 0.22 | 0.20 | 0.13 | 0.11 | 0.061 | 0.039 |
| | RSD (%) | 0.58 | 0.58 | 1.0 | 0.58 | 0.32 | 0.46 |
| | 900 | 0.22 | 0.18 | 0.15 | 0.11 | 0.071 | 0.047 |
| | | 0.21 | 0.19 | 0.16 | 0.12 | 0.075 | 0.041 |
| | | 0.22 | 0.18 | 0.14 | 0.11 | 0.081 | 0.048 |
| | RSD (%) | 0.58 | 0.58 | 1.0 | 0.58 | 0.5 | 0.38 |

## Chronic dietary risk assessment

A safety interval of 28 days after dimethoate application is recommended based on the MRL of omethoate in Chinese celery. The dietary intake risk was not calculated at different

**Table 7 Chronic risk quotient (RQ) of total residualof dimethoate and omethoate (expressed as dimethoate) of different populations in China.**

| Age (year)/sex | Body weight (kg) | Vegetable intake (F) (g d$^{-1}$) | Median residue (STMR)[*] (mg kg$^{-1}$) | | | chronic risk quotient (RQ) | | |
|---|---|---|---|---|---|---|---|---|
| | | | 10 d | 14 d | 21 d | 10 d | 14 d | 21 d |
| 2–7/irrespective | 17.9 | 194.8 | 0.73 | 0.26 | 0.15 | 3.97 | 1.41 | 0.82 |
| 8–12/irrespective | 33.1 | 272.4 | 0.73 | 0.26 | 0.15 | 3.00 | 1.07 | 0.62 |
| 13–19/male | 56.4 | 396.7 | 0.73 | 0.26 | 0.15 | 2.57 | 0.91 | 0.53 |
| 13–19/female | 50.0 | 317.9 | 0.73 | 0.26 | 0.15 | 2.32 | 0.83 | 0.48 |
| 20–50/male | 63.0 | 436.4 | 0.73 | 0.26 | 0.15 | 2.53 | 0.90 | 0.52 |
| 20–50/female | 56.0 | 412.1 | 0.73 | 0.26 | 0.15 | 2.69 | 0.96 | 0.55 |
| 51–65/male | 65.0 | 477.9 | 0.73 | 0.26 | 0.15 | 2.68 | 0.96 | 0.55 |
| 51–65/female | 58.0 | 447.0 | 0.73 | 0.26 | 0.15 | 2.81 | 1.00 | 0.58 |
| >65/male | 59.5 | 413.3 | 0.73 | 0.26 | 0.15 | 2.54 | 0.90 | 0.52 |
| >65/female | 52.0 | 364.1 | 0.73 | 0.26 | 0.15 | 2.56 | 0.91 | 0.53 |

Note:
[*] STMR is the median residue of dimethoate (sum of dimethoate and 3[*]omethoate, expresses as dimethoate) of the standard test in Tables 5 and 6.

times in this study. The standard trial median residue of the total of dimethoate and omethoate in celery is shown in Table 7. The allowable daily intake of dimethoate is 0.002 mg kg$^{-1}$ bw (GB 2763-2016, 2016). The daily consumption of vegetables is estimated based on the Chinese dietary structure (Wu, Zhao & Li, 2018; Liu et al., 2018). The daily intake of celery is lower than the total vegetable intake. Suppose the daily total vegetable intake replaces the celery intake. In that case the calculated dietary risk of the total residual of dimethoate and omethoate is acceptable in vegetable. The dietary risk of the total residual of dimethoate and omethoate in celery is acceptable.

The risk quotient (RQ) was calculated using the chronic dietary risk formulas 3 and 4. The results (Table 7) showed that on day 10, the RQs of dimethoate were both more than one, and therefore, the risks were unacceptable. On day 14, some RQs of dimethoate were more than 1 (2–12 years and 51–65 years/female), and the risks were unacceptable. After day 21, the RQs of dimethoate in celery were less than one, and the risk was acceptable.

## Acute dietary risk assessment

The acute reference dosages (ARfD) of dimethoate is 0.01 mg kg$^{-1}$ bw (Geng et al., 2018; Utture et al., 2012). The HQ was calculated based on the dietary structures of different populations in China (Wu, Zhao & Li, 2018) using the acute dietary risk assessment formulas 5 and 6 to judge the level of acute dietary risk (Table 8). The results showed that the HQ range of dimethoate was 2.42–4.15 on day 10. On day 14, the HQ of dimethoate was 1.22–2.09. On day 21, the HQ of dimethoate was 0.67–1.14. On days 10 and 14, the HQs of dimethoate were more than one and the acute risks were unacceptable. On day 21, only children aged 2–7 years had an HQ of dimethoate more than 1 (an intolerable level of risk), while the acute dietary threats to other populations were within acceptable levels. As a precaution, it was recommended that large amounts of single

**Table 8 Acute risk quotient (HQ) of total residualof dimethoate and omethoate (expressed as dimethoate) of different populations in China.**

| Age (year)/sex | Body weight (kg) | Vegetable intake (F) (g d$^{-1}$) | Highest residue (HR)[*] (mg kg$^{-1}$) | | | acute risk quotient (HQ) | | |
|---|---|---|---|---|---|---|---|---|
| | | | 10 d | 14 d | 21 d | 10 d | 14 d | 21 d |
| 2–7/irrespective | 17.9 | 194.8 | 1.27 | 0.64 | 0.35 | 4.15 | 2.09 | 1.14 |
| 8–12/irrespective | 33.1 | 272.4 | 1.27 | 0.64 | 0.35 | 3.14 | 1.58 | 0.86 |
| 13–19/male | 56.4 | 396.7 | 1.27 | 0.64 | 0.35 | 2.68 | 1.35 | 0.74 |
| 13–19/female | 50.0 | 317.9 | 1.27 | 0.64 | 0.35 | 2.42 | 1.22 | 0.67 |
| 20–50/male | 63.0 | 436.4 | 1.27 | 0.64 | 0.35 | 2.64 | 1.33 | 0.73 |
| 20–50/female | 56.0 | 412.1 | 1.27 | 0.64 | 0.35 | 2.80 | 1.41 | 0.77 |
| 51–65/male | 65.0 | 477.9 | 1.27 | 0.64 | 0.35 | 2.80 | 1.41 | 0.77 |
| 51–65/female | 58.0 | 447.0 | 1.27 | 0.64 | 0.35 | 2.94 | 1.48 | 0.81 |
| >65/male | 59.5 | 413.3 | 1.27 | 0.64 | 0.35 | 2.65 | 1.33 | 0.73 |
| >65/female | 52.0 | 364.1 | 1.27 | 0.64 | 0.35 | 2.67 | 1.34 | 0.74 |

**Note:**
[*] HR is the highest residue of dimethoate (sum of dimethoate and 6[*]omethoate, expresses as dimethoate) of the standard test in Tables 5 and 6.

types of food should not be included in for children aged 2–7 years in the short term to reduce acute dietary risk.

## DISCUSSION

This study found that the dissipation of dimethoate and total residues in greenhouse celery conformed to the first-order kinetic equation, with $r^2$ equal to 0.9943 and 0.9814, respectively, and half-lives of 2.42 days and 2.92 days, respectively.

Previous studies found that the half-life of dimethoate in the open field is 2.5 days (*Guo et al., 2017*), indicating that the residual level of dimethoate is not significantly different in the greenhouse and in the field. The half-life of dimethoate in mango is 2 days (*Bhattacherjee & Dikshit, 2016*), and the half-life in cucumber is 5.2 days (*Geng et al., 2018*), indicating that the dissipation of dimethoate is related to the matrix on which it is applied. The half-life of dimethoate in celery grown in Guizhou is 5.4 days, and the half-life of celery in Anhui is 3.5 days (*Chen et al., 2018*). Nevertheless, the dissipation of dimethoate is faster in Liaoning (85%) than in Guizhou (75%) and Anhui (70.27%) 7 days after application, indicating that the half-life of dimethoate is also related to region and climate.

As shown in the final residue results, spraying dimethoate's safety risks in the seedling, transplanting, and moderate growth stages were within acceptable limits. Specifically, the residue of dimethoate dropped to less than its MRL 14 days after the last application in the harvesting stage. Still, the residue of omethoate remained far higher than its MRL. Hence, dimethoate application's safety interval should be at least 28 days, which is similar to the respective safety intervals of 27 days for cucumber (*Geng et al., 2018*) and 30 days for pomegranate (*Utture et al., 2012*).

The results of the dietary risk assessments showed that after day 21, the RQs of dimethoate in celery were less than one and so the level of chronic risk was acceptable. Only children aged 2–7 years had an HQ of dimethoate more than 1 (an unacceptable level

of acute risk), while the acute dietary risks to other populations were within acceptable levels. Furthermore, from a toxicology perspective, the celery would be safe to eat at this safety interval even if the residual concentration of omethoate in celery was higher than the corresponding MRL. Poland and France have made similar assessments of exposure risks for dimethoate and omethoate in other foods (*Nougadère et al., 2014*; *Paweł et al., 2015*).

## CONCLUSION

This study showed that the application of dimethoate to greenhouse-grown celery resulted in omethoate residues exceeding the acceptable levels. Any applications of dimethoate to celery in greenhouses should be performed with a 28-day safety interval that ensures adequate levels of omethoate. As a precaution, it is recommended that large amounts of single types of food should be avoided in diets for children aged 2–7 years to reduce their dietary risk. This finding provided data to support the risk assessments of dimethoate and omethoate in celery and other foods. Although the standard residue test in this study was conducted in the Liaoning district, it provided the basis for testing in other regions of northern China. More importantly, the multiple-year residual data in many places may be combined to make these assessments more accurate.

### Funding

This work was supported by the Natural Science Foundation of Liaoning Province (No. 20170540502) and the Ministry of Agriculture and Rural Risk Assessment Program (No. GJFP2019008). The funders had no role in study design, data collection and analysis, decision to publish, or preparation of the manuscript.

### Grant Disclosures

The following grant information was disclosed by the authors:
Natural Science Foundation of Liaoning Province: 20170540502.
Ministry of Agriculture and Rural Risk Assessment Program: GJFP2019008.

### Competing Interests

The authors declare that they have no competing interests.

### Author Contributions

- Chunjing Guo conceived and designed the experiments, performed the experiments, prepared figures and/or tables, authored or reviewed drafts of the paper, and approved the final draft.
- Guang Li performed the experiments, analyzed the data, prepared figures and/or tables, and approved the final draft.
- Qiujun Lin analyzed the data, authored or reviewed drafts of the paper, and approved the final draft.

- Xianxin Wu analyzed the data, authored or reviewed drafts of the paper, and approved the final draft.
- Jianzhong Wang conceived and designed the experiments, prepared figures and/or tables, authored or reviewed drafts of the paper, and approved the final draft.

## Data Availability

All data are available in Tables 1–8.

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
