# Peer review of "Residual dynamics and dietary exposure risk of dimethoate and its metabolite in greenhouse celery"

_PeerJ, doi:10.7717/peerj.10789_

## Round 0.1 · original submission · Major Revisions

Please take into consideration the reviewer's comments, and provide a revised manuscript and a detailed point-by-point rebuttal letter.

I concur with the reviewer 1 that the study is questionable without presenting sufficient evidence of the reproducibility of the experiments.

Reviewer 1 ·

Basic reporting

The language of the manuscript is good. The introduction is sufficiently detailed, However, there is a contradiction in it namely the statement on dimethoate residues seriously affected human health. This statement has no ground and contradicts the conclusion. It should be removed or modified
The manuscript is well structured, the figures and tables are technically appropriate, however the values reported are questionable.
The topic selected is actual and relevant, however the reporting the experiments and the analyses is not providing the necessary details to judge the reality of the results.
Detailed comments are given in the manuscript.

Experimental design

Dimethoate 40EC was used for treating the crops. At day 0 0.16-0.18 mg/kg omethoate was detected. It warrants to check the purity of 40EC formulation and report its omethoate content
Pesticide application: specify application equipment, its calibration, and report the actual dose delivered.
Pls specify the time interval required for celery to reach mature stage form seedling, planting and middle growth stage.
Check the registered use pattern of celery. Applying 20 times higher dosage than recommended in EU seems questionable.
Sampling: specify the portion of commodity sampled, report the number of primary sampling position from which the 2 kg laboratory sample was collected at each sampling time
How long the harvesting period lasts? What is the practical relevance of applying the pesticide 2-3 times at 7 days interval? What is the relevance of such application in view of the official preharvest interval?
What is the time between middle growth stage and reaching the maturity of celery?

Laboratory sampling operations: specify the part of the mature celery analyzed, the portion of the laboratory sample comminuted and the equipment used.
Analysis: what is the reproducibility of determination of the compounds? Recovery studies does not reveal any information on the reproducibility of analyses on which the decline studies were based.
Describe performance of recovery tests. At day 0 about 4 mg/kg after 3 days 1.7-2.0 mg/kg residues was measured??? The highest level of recovery test was 1 mg/kg.
The LOQ, LOD of the method should be reported.
There is no point to compare average of residues measured in three replicate samples. Residues in each sample should comply with MRL.
Exposure assessment. It is not clear what was the basis and relevance of estimating median residues for dietary risk assessment.
What was the reason to disregard median residue in equ. 5 considering the consumption patterns reported?

Validity of the findings

Based on the reported data the validity of the results and conclusions cannot be judged. See detailed comments above

Additional comments

In summary: based on the applied experimental design and analyses of samples reported, the validity of the study cannot be judged, and the credibility of the conclusions is rather questionable.
If the missing information is available for the authors and the experimental part of the study can be properly reported addressing the above missing points, then a major revision of the manuscript might be worthwhile. Otherwise the manuscript is not suitable for publication.

Annotated reviews are not available for download in order to protect the identity of reviewers who chose to remain anonymous.

Reviewer 2 ·

Basic reporting

no comment

Experimental design

no comment

Validity of the findings

no comment

Additional comments

The study investigates a relevant issue and is well writen and professional.
Please see minor corrrections and suggestions regarding the conclusions to be emphasized in the atteched pdf document in the comments.
Congratulations and best regards.

Annotated reviews are not available for download in order to protect the identity of reviewers who chose to remain anonymous.

---

## Round 0.2 · Major Revisions

Please attend the detailed comments of the reviewer, and present a strong case in a detailed rebuttal letter.

Reviewer 1 ·

Basic reporting

The authors tried to improve the description of experiments, but there are several major discrepancies which have to be sorted out before the paper can be accepted for publication.

Experimental design

The text does not provide clear information on the implementation of pesticide applications and sampling days

Validity of the findings

It cannot be judged before the experimental design is clarified and precisely described.

Additional comments

General comment: Please clearly link growing stage, days of pesticide applications, days between last pesticide treatment and sampling and residues measured. Perhaps a graphical presentation would help to make the experimental design and it description unambiguous.
Specific comments see also attached manuscript with tracked changes
L. 19-27 Since omethoate derives from dimethoate, there is no point in reporting residues after 28 days, when the PHI for dimethoate is 14 days. What is the justification for calculation of acute intake after 21 days and chronic intake after 28 days? Pls. state clearly what is the registered PHI for dimethoate in celery and perform the calculations for that (or shorter) interval.
L. 39. the problem here is not 550,000, but the use of “h” (hecto) instead of internationally accepted units. The description of application rate above is also confusing and should be corrected.
L.133. How many days after last pesticide application was the sampling performed? How pesticide applications were made before ’one time point sampling’? Please clearly separate the trials with sequential sampling and ’one time point sampling.
L243-245: pls. revise the sentence: state MRL of dimethoate and compare the residues measured to the MRL. Dissipation rate has no relation to MRL.
L.258. Pls. reconsider this conclusion: If the Day 0 samples were taken shortly after the first application of dimethoate, the omethoate concentration should be practically close to LOQ, then gradually increase as the dimethoate degrades. So it is a complex time-conc. relationship and describing it as 1st order reaction is a very rough estimation. Probably from day 3 the decline could be described as 1st order.
L. 269 Reconsider Table 4. Clearly indicate the days of pesticide application and the growing stage of crop together with the residues measured
L. 273. This paragraph does not reflect the table. Both should be reconsidered and revised accordingly.
Table 5. Indicate the days of pesticide treatment in connection with sampling days
Table 6. The title of the table and it content do not match
Table 7 To consider total vegetable intake (instead of celery) provides a very uncertain result because there is no information on the residue content of other vegetable consumed on the same day. This uncertainty shall be pointed out in the article
Discussion section cannot be evaluated until the elucidation of unclear points indicated above
L.368. perhaps: would lead to omethoate residues which would not risk the health of consumers
L. 373 It was indicated that the dissipation rate of dimethoate and omethoate residues is affected by several factors. What is the basis of such a general statement?

Annotated reviews are not available for download in order to protect the identity of reviewers who chose to remain anonymous.

---

## Round 0.3 · Major Revisions

Please take into consideration the reviewer’s comments and provide back a point-by-point rebuttal letter addressing those concerns.
In particular, is critical that the tracked changes version is the same as the accepted changes. Also, the risk assessment section needs to fully consider the comments of reviewer 1.

Reviewer 1 ·

Basic reporting

The write-up has been improved but still there are a number of points to be corrected. They are indicated in the annotated manuscript.
No attempt was made to edit the language.

Experimental design

It has been clarified and now the experiment results can be evaluated

Validity of the findings

The analytical part is marginally acceptable. However, the risk assessment would require full revision.

Additional comments

The word document with tracked changes do not correspond with the pdf document. The applied dosage is not clear. Is it 15 g formulated product per 100 m2 or 15 g active substance/100 m2
Abstracts and text hectare is an area (10,000 square meter) it should not be put on second power ha2
L. 80. specify dosage in g active ingredient (ai.) or formulation.
Repeatability of recovery tests does not reflect the uncertainty of reported residue values. To determine the uncertainty of reported residue values several test portions should be taken from blended material, and the relative standard deviation of the replicate measurements of residues in treated celery should be calculated and reported together with the measured residue values.

Risk assessment is incorrect as the sum of residues of dimethoate and X×omethoate should be taken into account (X is to account for the higher toxicity of omethoate). Note that the toxicological endpoints are the same so the parent compound and its metabolite should be considered together for both acute and chronic dietary intake (for instance see Tables 5 and 6)

Figure 3. It is not correct to describe omethoate dissipation as 1st order reaction including the first 3 days (at day 0 omethoate concentration should be below LOD). The explanation provided is not sufficient for supporting the calculation presented.
Table 4. sub-title is misleading: just write seedling stage and place a footnote: final residues following treatment at seedling stage; same comments for other sub-titles
Table 7: the calculation of STMR values should be explained in footnotes or in the text. As indicated above the sum of dimethoate and omethoate shall be considered.
Moreover, please note that MRLs are not safety limits and should not be used as reference point for evaluation of dietary risk.
Table 7 and 8: why the STMR values are different in the tables?

Annotated reviews are not available for download in order to protect the identity of reviewers who chose to remain anonymous.

Reviewer 2 ·

Basic reporting

No comment

Experimental design

No comment

Validity of the findings

No comment

Additional comments

The study investigates a relevant issue and is well written and professional.
Unfortunately, some more corrections are needed for tha manuscript to be accepted. Please see suggestions for minor corrections in the attached pdf document in the comments.

Annotated reviews are not available for download in order to protect the identity of reviewers who chose to remain anonymous.

---

## Round 0.4 · Minor Revisions

I have included some suggestions in a PDF file since there are still issues in English grammar and style. Please obtain a thorough revision from an English speaker copywriter or a professional edition service.

n

---

## Round 0.5 · Minor Revisions

The manuscript has improved, please attend the following comments:

In general, the practice of pesticide application appears routine; might there be some mention of other growth regimes where pesticides are not used or available?

As the manuscript focuses on the dimethoate EC pesticide it would be useful to include illustrations of the chemical makeup formula and its break down products.

One might ask that there should be some pointer to the Pesticide Fact Sheet because this is a chemical applied to a food product to inform the reader of the toxicity of the compound on food as the chemical is the key subject matter, and to clarify more that this is a standard practice in treatment that ends in a product for human consumption somewhere.

SUGGESTED EDITS:
LINE NO: / BEFORE / AFTER / [COMMENTS]
LINE 27: / who / which / [.]
LINE 112: / housr / hours / [.]"

---

## Round 0.6 · accepted · Accept

Thanks for addressing all the revisions and corrections requested. Now your manuscript is accepted in PeerJ.